# A Dynamic Federated Identity Management Using OpenID Connect

Ahmad Alsadeh [1,*], Nasri Yatim [2] and Yousef Hassouneh [2]

1   Electrical and Computer Engineering Department, Birzeit University, Birzeit P.O. Box 14, Palestine
2   Computer Science Department, Birzeit University, Birzeit P.O. Box 14, Palestine
*   Correspondence: asadeh@birzeit.edu

**Abstract:** Identity federation allows one to link a user's digital identities across several identity management systems. Federated identity management (FIM) ensures that users have easy access to the available resources. However, scaling FIM to numerous partners is a challenging process due to the interoperability issue between different federation architectures. This study proposes a dynamic identity federation model to eliminate the manual configuration steps needed to establish an organizational identity federation by utilizing the OpenID Connect (OIDC) framework. The proposed model consists of three major steps to establish dynamic FIM: first, the discovery of the OpenID service provider, which indicates the location of the partner organization; second, the registration of the OpenID relying party, which allows the organization and its partner to negotiate information for establishing the federation; finally, establishing the dynamic trust federation. The proposed dynamic FIM model allows applications to provide services to end-users coming from various domains while maintaining a trust between clients and service providers. Through our proposed dynamic identity federation model, organizations can save hundreds of hours by achieving dynamic federation in runtime and serving a large number of end-users.

**Keywords:** identity management; identity federation; OpenID connect; dynamic client registration

## 1. Introduction

The growth in the use of business outsourcing and collaborative platforms causes demand for organizations to share the identity information they maintain about their users with other partners. Collaboration and sensitive data sharing are protected by the legislation of the organizations' countries. However, the sensitive data should not be transmitted on the Internet insecurely since it creates security and privacy risks. Organizations adhere to federated environments using a federated identity management (FIM) standard to cope with this growth and to add value to the business by allowing third parties services. The literature in the past decade has been proposing different approaches for establishing trust federation. Many problems and concerns might arise when it comes to managing and securing users' accounts, identifiers and passwords in a highly dynamic and insecure environments. Therefore, establishing an identity trust federation between an organization and another entity removes the complexity and concerns regarding identity management for organizations, allowing them to focus on providing the services they want and delegate the identity management tasks to specialized entities.

The state of identity management today relies on federation protocols, such as the Security Assertion Markup Language (SAML), Open Authentication 2.0 (OAuth 2.0) and OpenID Connect (OIDC) for external communication of identity credentials. SAML allows the exchange of user authentication and information using XML between different domains [1], OAuth 2.0 allows users to give applications permissions to access resources on their behalf [2], and OIDC allows identity to be communicated in a RESTful-like manner [3].

Identity federation allows one to link a user's digital identities across several identity management systems. When an organization needs to collaborate with a partner's platform,

the organization does not release a copy of the user's store (user's credentials) in order to allow the user to authenticate with the partner's platform. Instead, the organization provides an identity management system to the partner's platform to utilize the stored identities, given that the organization trusts the business partner's platform. Therefore, an identity federation protocol allows the decoupling of the authentication and the authorization functions, removes the security risk for managing multiple credentials by the organization users and creates a safe channel for identities to be shared across different domains.

The adoption of IoT solutions, such as smart-home and e-health, raises a new security concern that requires providing an identity federation framework [4,5]. The information between multiple IoT devices should be seamlessly exchanged across all domains to achieve authentication and authorization. For example, in an e-health IoT service, users are able to use their single account for all the e-health provider's services for all the IoT devices via identity federation [4].

Although organizations gain significant business value using identity federation management techniques, they continue to face major obstacles especially when it comes to scaling up to hundreds of business partners [1]. Each business partner might use different identity federation architectures, which raises an issue in interoperability between those different federation architectures [6].

OIDC [3] is a common authentication protocol and its popularity has been increasing enormously since its launch in 2014. A large number of online applications are adopting OIDC protocol to allow them to request and receive information about the authenticated end-user without the hassle of managing end-users' identities. Users can see OIDC when a web applications allows them to sign in using Facebook, Google, Microsoft's Office 365 or other OPs. OIDC is considered one of the first options by software developers when it comes to application programmable interface's (API) security [7], since it provides:

- Easy consumption of identity tokens: an end-user's identity is received by client applications in a secure JSON Web Token (JWT) called the ID token. JWTs are portable and support a range of signature and encryption algorithms [7].
- OAuth 2.0 protocol: OIDC is built on top of OAuth 2.0, meaning that clients use OAuth 2.0's various authorization grant types to obtain ID tokens and access tokens, which work with both web applications and native mobile applications [7].
- Simplicity: OIDC is relatively simple to integrate with applications, while also offering features and security options that can meet demanding enterprise requirements [7].

Through OIDC, organizations can establish a federated environment with third-party providers in order to extend their services. However, the process of establishing a federated identity is a relatively complex task. The complexity comes from the rapid need of businesses collaborations and outsourcing.

This study addresses the following research question: How can a dynamic OIDC federated identity model be applied to assist organizations to scale-up to a large number of business partners? To answer this question, we propose a model that is able to achieve a dynamic identity federation between two entities leveraging OIDC. The model is based on three major OIDC specifications [8–10]. The model utilizes those specifications to allow OIDC relying parties to locate an end-user's identity providers, register with them as an OIDC relying party (RP) and establish a trust through public key infrastructure (PKI) in order to obtain the needed information for authenticating an end-user and achieving identity federation.

*Contribution*

This study provides a model that enables organizations to dynamically establish a federated identity management (FIM) framework between each other to cope with the increase needs of business collaboration. The proposed model allows organizations to serve end-users coming from any domain, even if the organization does not recognise this domain, since it establishes the identity federation and trust dynamically and at runtime. The proposed model consists of three major processes necessary for establishing

the identity federation: dynamic discovery, dynamic client registration and dynamic trust federation establishment.

## 2. Related Work

Wang et al. [11] proposed an approach to achieve unified identity authentication using Microsoft Active Directory (AD). The authors argued that with the growth of enterprise systems and with minimal planning for unified solutions, each new system integrated with the enterprise is deployed separately and manages a different set of identities. Therefore, users of these multiple systems have to manage multiple credentials to access each system, which leads to some users experiencing issues along with security and maintainability issues. In such cases, usually a single sign-on (SSO) solution is needed, which acts as a centralized authentication system that allows users of the enterprise to sign in only once as their identities is being managed by a single entity. The main goal of the paper is to integrate AD authentication and validation to the application system in order to achieve a unified management authorization service that can realize cross-platform and cross-language system applications.

Jian et al. [12] based their research on establishing a trust relationship between two communicating parties that need to trust each other in a nontrusted environment. Establishing this trust relationship allows users of one organization to access servers of the other organization securely and seamlessly. They proposed a new component called trust server provider (TSP) that acted as a middleware third party to manage the trust relationship between the two communicating parties. This was done by requiring every communicating party to register their information with the TSP and obtain a public key. The TSP then could help the communication between parties by establishing a new trust relationship at runtime. The TSP also supported updating parties' information and revoking their public keys, if any of the parties needed to not be trusted any more. The main contribution of their study was the establishment of a trust framework that allowed communicating parties to trust each other and achieve a unified single sign-on using SAML.

Harding et al. [1] proposed a dynamic SAML approach that aimed to automate the exchange of configuration information and to minimize manual steps necessary to establish a trust federation between business partners to allow identity federation and single sign-on (SSO). The main goal of the paper was to cope with the continuous growth of business collaboration platforms that require businesses to share the identity information of their users with their partners. SAML by its nature is designed to deliver SSO and other security attributes that provide organizations techniques for identity federation management. Although SAML provides great security, its deployment time is a hurdle for organizations, and deploying SAML-based projects may take weeks or months because of the lack of standardized mechanisms for metadata exchange and trust establishment. Through dynamic SAML, the authors were able to establish three main goals. The first goal was to create and maintain a trust framework between hundreds of business partners with the least minimal resources through cryptography and X509 certificates. The second goal was to create a dynamic SAML document metadata exchange that allowed SAML entities to establish an SSO connection instantly, allowing the least minimal resources to be used for SAML deployment. The third and final goal was to secure the SAML metadata exchange by guaranteeing metadata authenticity through a digital signature. As a result, dynamic SAML both simplified the process of deploying federation technologies and ensured secure deployments.

Bendiab et al. [13] proposed a trust model that aimed to integrate identity federation systems into the cloud environment. The model leveraged the fuzzy cognitive maps (FCM) tool to establish, model and evaluate a trust relationship between two entities. Upon end-user authentication, the RP received a signed access token and used it to access a protected resource from the resource server. When the RP provided the token and sent an API call to request the resource, the resource server (or the API) computed the trustworthiness level, leveraging the FCM tool of the identity provider (IdP) from which the token was issued.

If the value of trustworthiness was above a certain threshold, the IdP could be trusted and resource access could be granted. The motivation behind this model was that existing identity federation models, such as SAML and OIDC, were limited by the complexity of their trust model. This complexity comes from the fact that these models are based on a preconfigured circle of trust (CoT), which is hard to scale and is not extendable. In addition, this model provided a confidence level in a user's identity trust, unlike OIDC which lacks that confidence.

Ferdous et al. [14] proposed a model based on a drafted SAML profile that allowed users to establish a trust federation dynamically between two organizations, resolving the limiting dynamic federation support in the current SAML implementation. The author claimed that the proposed dynamic SAML model allowed users to create the identity federation dynamically and in real-time with a specific lifetime threshold, meaning that the federation is removed once the threshold has passed.

Korse et al. [15] presented a multifactor authentication mechanism named TrustedID system that was based on Mobile Connect and OpenID Connect standards to access sensitive mobile services on a smartphone.

Bendiab et al. [16] introduced a trust and identity management model based on the blockchain for cloud identity management. In another study, Yang and Li [17] leveraged the smart contracts and zero-knowledge proof (ZKP) algorithms to avoid the threats that might arise by using a centralized third party for improving the existing claim identity model in a blockchain. Moreover, Mell et al. [18] used a smart contract on a blockchain to eliminate the need for a third party management while enabling users to maintain a level of self-sovereignty.

Our literature review indicated that no research until this time provided a dynamic identity federation model using OIDC. In fact, Bendiab et al. [13] in 2018, and others [14], stated that the OIDC registration process is hard to scale and not technically extendable, OIDC does not provide a protocol that specifies a trust model to establish a trust relationship between two entities, and OIDC has limitations and cannot be deployed in a dynamic and open environment, such as the cloud.

In mid 2019 until September 2021, the OpenID Foundation issued a new identity federation trust protocol named OIDC Federation 1.0 [10]. The OIDC protocol specified how an organization could dynamically discover and register with an identity provider by establishing a federated identity model along with establishing a trust model between those two entities. We believe that the recent release of the OIDC federation protocol introduced a new opportunity to utilize OIDC for dynamic identity federation.

### 3. The Proposed Approach

To answer the research question and to address the challenge of scaling up organizations, a dynamic federated identity model is proposed by leveraging the OIDC framework to provide organizations the capability to dynamically scale up their businesses and establish an identity federation management framework. The approach defines several models that can be combined and used to achieve a dynamic federation between organizations. As a result, end-users accessing any web-enabled services should be able to use a single identity while accessing all those services. The proposed model is implemented as a proof-of-concept in a controlled environment to help us answer the research question.

Our identity provider (IdP) was Microsoft Active Directory Federation Services (ADFS), since it supports OIDC authentication. An authorization server proxy was developed to make ADFS support dynamic client registration. Other identity providers that support dynamic client registration could be used instead of ADFS (https://docs.microsoft.com/en-us/windows-server/identity/active-directory-federation-services, (accessed on 16 November 2022), such as IBM Cloud Identity (https://cloud.ibm.com/docs/account?topic=account-iamoverview, (accessed on 16 November 2022), Authelete 2.2 (https://www.authlete.com, (accessed on 16 November 2022), Cloudentity (https:

//cloudentity.com/why-cloudentity, (accessed on 16 November 2022) and Gluu Server 4.2 (https://gluu.org, (accessed on 16 November 2022).

The proposed model consisted of three major processes necessary for establishing identity federation: dynamic discovery, dynamic client registration and dynamic trust federation establishment.

1. Dynamic discovery: The dynamic discovery process answers this question to a service provider *"Where are end-users coming from?"*. This is necessary in order for an RP to know where and how to authorize end-users for granting them access to its services.
2. Dynamic client registration: In order for an RP to be able to use OIDC services, the RP needs to register with an OpenID provider (OP). The process allows RPs to dynamically register themselves by providing information about themselves and to obtain the necessary information.
3. Dynamic trust federation: The previous two steps do not define a mechanism to establishing a trust between OIDC RPs and OPs. The dynamic trust federation process allows RPs and OPs to dynamically obtain and establish a trust from a common trusted third party. Establishing this trust allows entities to trust the information communicated between each other. An entity can establish a cryptographically based trust relationship with other entities via a certificates exchange that anchors these trust relationships into a trust anchor list. The main purpose of a trust anchor entity is to issue certificates to other entities participating in the trust federation. This way, relying parties can only register with OPs by exchanging certificates in a signed statement whose certificate validates the signature against its trust anchor list.

## 4. Experimental Environment and Implementation

Our proposed model is presented along with its implementation details, which allows an organization to establish a dynamic federated identity environment with third-party providers using OIDC. Assume Bob works in a marketing and advertisement company called `AdvertiseMe`. Bob's supervisor asked Bob to create a marketing flyer for an event that will take place next month. Bob finds an online tool to help him create a marketing flyer named `FlyerIt`. However, `FlyerIt` requires users to log in to use all the features provided by this tool. Bob would like to sign in to `FlyerIt` using his company email address `bob@advertiseme.com` without registering as a new user and creating a new password. Using our proposed dynamic federation model, `FlyerIt` should allow Bob to use his `AdvertiseMe` company identity to sign in to the system even though it does not know anything about `AdvertiseMe`'s identity servers and how to authorize its users, and if it is even a trusted entity. `FlyerIt` should also be able to retrieve Bob's necessary user information from his `AdvertiseMe` company, such as his profile picture, name and company address. At the end, Bob uses his company email address and password, is granted permissions and is able to access all the features provided by the `FlyerIt` tool to create a flyer.

### 4.1. Dynamic OIDC Provider Discovery

We assume that `AdvertiseMe` is a company that has its users stored in ADFS and is capable of providing claims to clients or relying parties. `FlyerIT` is a web application that provides web services and its resources requires users to login using any of the supported identity providers. We assume that Bob has never used the `FlyerIt` tool before. If Bob wants to use his `AdvertiseMe` identity to log in to the tool, how would the tool know who is the identity provider of Bob to authenticate him? Dynamic discovery helps `FlyerIt` as an RP to dynamically allocate an end-user's identity provider. The discovery process is important so that the RP can obtain the OP information needed for interaction (e.g., registration and authorization) with the identity provider. The discovery process is the first process towards achieving a dynamic identity federation environment. This step is initiated only once. Once an RP has discovered the IdP of a specific host, the RP is able to

serve end-users coming from the same host indefinitely. Dynamic discovery is possible through the use of `WebFinger` [19] to locate the OP for a specified end-user.

Figure 1 depicts the sequence diagram that defines the steps for `FlyerIt` to achieve the dynamic discovery of `AdvertiseMe`.

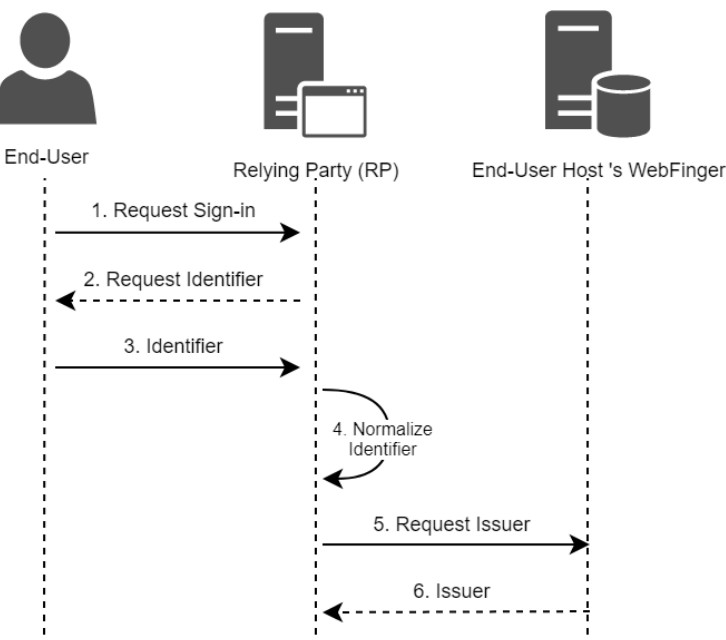

**Figure 1.** OIDC Dynamic Discovery Model.

1.  Dynamic discovery is triggered whenever a user requests a sign-in from an RP in order to access their resources and services. Bob is the initiator of this process when he triggers the `FlyerIt` RP for a custom sign-in using his company's credentials.
2.  The RP (`FlyerIt`) requests the end-user to provide an identifier, which is necessary so that the RP can extract the resource and the host. In our scenario, the RP requests Bob to provide an identifier.
3.  The end-user provides an identifier to the RP. The RP accepts an email address as an identifier. Now, Bob provides his email address `bob@advertiseme.com` to the RP (`FlyerIt`).
4.  The RP extracts the identifier provided by the end-user and applies a normalization in order to determine the resource and host values. `FlyerIt` extracts Bob's email and determines that the resource is equal to `Bob`, and the host is equal to `advertiseme.com`.
5.  The RP prepares a `WebFinger` request to the resource's host in order to discover and receive the issuer (IdP) location. In our scenario, `FlyerIt` prepares the following request. The `HTTP GET` request is sent to http://advertiseme.com/.well-known/webfinger (accessed on 16 November 2022). The `/.well-known/webfinger` path is a well-known path that is defined in the protocol and all OPs supporting dynamic client registration should implement this path to allow discovery. In addition, the path query `re-source=acct:bob@advertiseme.com` is presented to indicate the resource that is being discovered. The `acct` scheme should be attached at the beginning of the identifier [8]. In addition, the path query `rel=http://openid.net/specs/connect/1.0/issuer` indicates that we are requesting the issuer's location.

```
GET /.well-known/webfinger
?resource=acct:bob@advertiseme.com
&rel=http://openid.net/specs/connect/1.0/issuer
HTTP/1.1
Host: advertiseme.com
```

6. Once the OP receives the request, it figures out the issuer of the provided resource and returns it to the RP as a `WebFinger` response as depicted below. The `WebFinger` response contains a `links` array that consists of a `rel` and `href` values for the specified subject. The `rel` indicates the type of location returned, which is an issuer location in our case. The `href` is the *URL* of the specified issuer.

```
HTTP/1.1 200 OK
Content-Type: application/jrd+json
{
"subject": "acct:bob@advertiseme.com",
"links":
[{
"rel": "http://openid.net/specs/connect/1.0/issuer",
"href": "https://identity.advertiseme.com"
}]
}
```

As a result of the steps defined previously, the RP has successfully discovered the location of the IdP of the end-user. `FlyerIt`, in our case, has discovered the locations of Bob's OP. As a result, `FlyerIt` is able to register itself with `AdvertiseMe` and authenticate Bob to access its resources.

## 4.2. Dynamic OIDC RP Registration

In order for Bob to be able to sign in to `FlyerIt` using his company's credentials, `FlyerIt` should dynamically register with Bob company's `AdvertiseMe` IdP in order to complete the login. Before initiating the OAuth 2.0 protocol, and in order to utilize all services for an end-user, the RP should register with the OP. The process of dynamic RP registration (known as dynamic client registration) is a protocol defined by both OAuth 2.0 [20] and OIDC [9]. The main concern is to allow RPs to register with the OPs dynamically at runtime to allow OAuth 2.0 authorization and OIDC authentication.

Figure 2 illustrates the abstract flow of the dynamic client registration process between an RP (or client) and an OP (or authorization server).

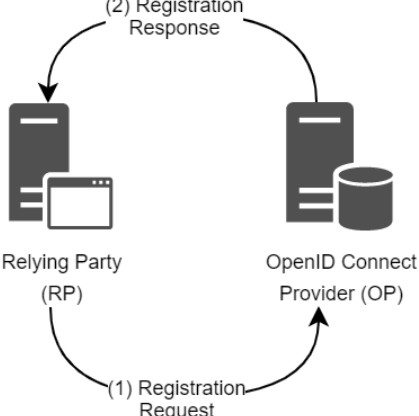

**Figure 2.** OIDC Dynamic Client Registration.

1. To register a new client, the client sends a registration request, which is an `HTTP` message to the authorization server with the important client metadata parameters.
2. The authorization server validates the client metadata and assigns a new client with the provided metadata in the request parameters. Then, the authorization server sends a registration response, which is also an `HTTP` message. The registration response contains information about the newly created client. It also sometimes includes a client secret that allows the client to authenticate itself while initiating an OAuth 2.0 authorization flow. The registration response depends solely on the information provided by the client.

Figure 2 shows back and forth communication between the RP and the identity provider. The nature of this communication might vary from one identity provider to another. Some identity providers require more security on their registration endpoints. Our proposed model for dynamic client registration requires all registration endpoints to be over TLS. In addition, we add an extra layer of security that allows the identity provider to establish a trust with the RP. This trust is important to establish a federation between the two entities. We discuss our trust establishment model in Section 4.3. For now, we illustrate the basic communication between an RP and an identity provider to enable dynamic client registration.

Figure 3 illustrates the complete flow of dynamic client registration;

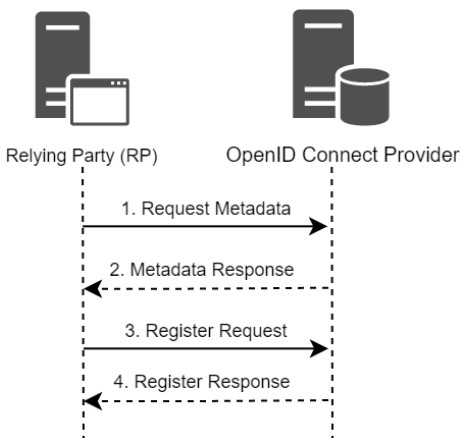

**Figure 3.** OIDC Dynamic Client Registration Model.

1.　The dynamic discovery process allows `FlyerIt` as an RP to identify the location of the IdP for `AdvertiseMe`. After knowing the location, `FlyerIt` attempts to request metadata information of `AdvertiseMe` via a configuration endpoint defined by `AdvertiseMe`. OIDC defines a configuration endpoint `/.well-known/openid-configuration` that must be implemented in all OPs, to allow clients to retrieve information about OPs including the registration endpoints. Thus, `FlyerIt` as an RP prepares the following HTTP request.

```
GET /.well-known/openid-configuration
HTTP/1.1
Host: advertiseme.com
```

2.　After `FlyerIt` sends the configuration metadata request, OP `AdvertiseMe` returns all metadata configuration information in a metadata response in terms of a JSON response. The configuration metadata contain a `registration endpoint`, which is the registration URL that allows clients to register with the server. Other metadata, such as the `authorization endpoint` and `token endpoint`, are URLs needed for end-user authorization and `access_token` retrieval. In addition, the OP server also provides a JWKS URL, which contains all the signing keys used to sign access tokens and identity tokens. Providing the signing keys allows relying parties to validate the integrity and authenticity of tokens created by the OP server.

```
{
"issuer": "https://identity.advertiseme.com",
"authorization_endpoint": "https://identity.advertiseme.com/oauth2/v2/auth",
"device_authorization_endpoint": "https://identity.advertiseme.com/code",
"token_endpoint": "https://identity.advertiseme.com/token",
"registration_endpoint": "https://identity.advertiseme.com/register",
"userinfo_endpoint": "https://identity.advertiseme.com/v1/userinfo",
"revocation_endpoint": "https://oauth2.advertiseme.com/revoke",
"jwks_uri": "https://identity.advertiseme.com/oauth2/v3/certs",
"response_types_supported": [
```

```
"code",
"token",
"id_token",
"code token",
"code id_token",
"token id_token",
"code token id_token",
"none"
],
"subject_types_supported": [
"public"
],
"id_token_signing_alg_values_supported": [
"RS256"
],
"scopes_supported": [
"openid",
"email",
"profile"
],
"token_endpoint_auth_methods_supported": [
"client_secret_post",
"client_secret_basic"
],
"claims_supported": [
"aud",
"email",
"email_verified",
"exp",
"family_name",
"given_name",
"iat",
"iss",
"locale",
"name",
"picture",
"sub"
],
"code_challenge_methods_supported": [
"plain",
"S256"
],
"grant_types_supported": [
"authorization_code",
"refresh_token",
"urn:ietf:params:oauth:grant-type:device_code",
"urn:ietf:params:oauth:grant-type:jwt-bearer"
]
}
}
```

3. Now that the `FlyerIt` RP has identified the registration `URL`, it prepares the registration request to add itself as a client for the OP `AdvertiseMe`. The registration request is an `HTTP` request to the registration endpoint `URL` containing client metadata parameters that the client uses in order to identify itself to the authorization server. An example `HTTP POST` request sent from the `FlyerIt` RP to the `AdvertiseMe` OP is as follows. The registration request contains the following parameters:

  - `redirect_uris`: The `redirect_uris` are the locations where an authorization server sends the end-user after the end-user authorizes the client applications.

In our scenario, `FlyerIt` provides those URIs so that `AdvertiseMe` can send Bob back to those URIs after Bob authorizes and grants `FlyerIt` permissions to access his basic profile information.

- `client_name`: A friendly name to be represented to the end-user so the user can identify the client.
- `grant_types`: OAuth 2.0 grant types are identified by the client to be used. Specifying these types will restrict the client to only using those types.

```
POST /register HTTP/1.1
Content-Type: application/json
Accept: application/json
Host: identity.advertiseme.com
{
"redirect_uris":
["https://flyerit.com/callback1",
"https://flyerit.com/callback2" ],
"client_name": "FlyerIt Forever",
"grant_types":
["authorization_code",
"client_credentials"],
"response_types": ["code"]
}
```

4. After the client sends the registration request, the authorization server validates the request and returns the newly created client identifier along with a client secret if applicable and other registration metadata. As a result of dynamic client registration, the RP can utilize OIDC services to serve end-users. `FlyerIt` can now allow Bob to sign in and use all the services it provides using Bob's company ID. In addition, `FlyerIt` can retrieve basic user information about Bob. The following sample is a registration response returned by `AdvertiseMe`.

```
HTTP/1.1 201 Created
Content-Type: application/json
{
"client_id":
"f1e76c1d-88e2-413c-a287-99c320deef12",
"client_secret":
"RT8V4bHvZqkA4n6DRsPJEGFd6arTdAWFjHnkZgdM",
"redirect_uris":
["https://flyerit.com/callback1",
"https://flyerit.com/callback2" ],
"client_name": "FlyerIt Forever",
"grant_types":
["authorization_code",
"client_credentials"],
}
```

### 4.3. Dynamic OIDC Trust Framework

Although identity federation offers better services to organizations at a lower cost, it also proposes new security threats. These security threats allow malicious entities to use a user's information or web resources. OAuth 2.0, based on a comprehensive threat model developed in 2013 [21], documents additional security considerations that include attacks on OAuth 2.0 tokens and protected resources. However, when it comes to crossing security domains in a federated identity, as well as a dynamic approach to allow discovery and client registration, many more security concerns arise. Thus, the proposed model so far raises some security concerns and is vulnerable to security attacks, such as phishing, broken authentications, replay attacks, on-path attacks, session hijacking, SSRF and code injection.

To tackle the security concerns in our proposed model, we propose a new trust model based on the new OIDC federation protocol [10]. This model aims to establish

a trust relationship between RPs and OPs. The proposed trust model is built based on PKI. The main goal of this concept is to verify that an RP is the entity it is claiming to be. The proposed trust model first aims to enhance the dynamic discovery model by leveraging digital certificates and PKI. This allows RPs to retrieve OPs' digital certificates upon discovery and validate their integrity and trustworthiness by performing certificate path validation. Accordingly, the RPs are able to identify malicious authorization servers. Second, it aims to enhance the dynamic client registration process by also leveraging digital certificates, which allows a real-time trust assessment and establishment between RPs and OPs. The model forces RPs to create a digitally signed software statement as a registration request and provide it to the authorization server. This in turn allows authorization servers (or OPs) to validate the signature of the statement, prove the possession of a private key and validates the integrity of the client to be registered by evaluating its trust chain for the provided certificate. As a result, malicious RPs are not able to register themselves thus preventing security concerns, such as phishing attacks.

Figure 4 illustrates a high level overview of the enhanced model that leverages PKI and digital identities in both dynamic IdP discovery and dynamic client registration.

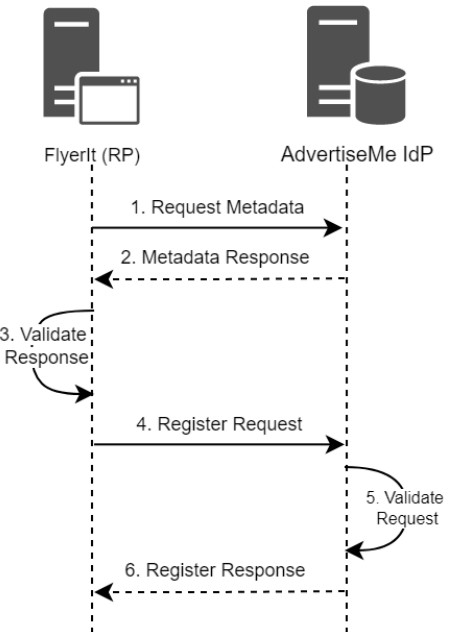

**Figure 4.** `FlyerIt` Dynamic Client Registration.

1.  Upon discovery, `FlyerIt` requests that authorization server's metadata information from a well-known configuration endpoint defined by `AdvertiseMe`.
2.  `AdvertiseMe` returns all metadata information as a JSON object.
3.  `FlyerIt` should not allow users to be able to authenticate themselves via a malicious organization's authorization server. The least it can do is to verify the identity and the integrity of this organization. Using PKI and certificate path validation, `FlyerIt` can verify the integrity, trustworthiness and appropriateness of the certificate provided by `AdvertiseMe`. `FlyerIt` now can extract the set of certificates provided in the JWKS claim. The claim contains a set of public keys that are used to verify tokens that are issued, signed and sometimes encrypted by the authorization server. The "x5c" claim contains the X.509 public key certificate or certificate chain corresponding to the key that is used by the authorization server (i.e., `AdvertiseMe`) to digitally sign tokens. The value of the "x5c" claim is a PEM encoded certificate, which is a block of encoded text that contains all of the certificate information and public key. Once the `FlyerIt` is in possession of the server's certificate, it can start building a PKI certificate chain from `AdvertiseMe`'s end-entity certificate up to the first root CA that `FlyerIt` trusts. If the verification is successful, `FlyerIt` can verify the certificate revocation list, or

CRL, published by the root CA. If the CRL does not indicate any revocation status, `AdvertiseMe` digital certificate is considered valid and can be relied upon.

```
{
"keys":
[
{
"kty": "RSA",
"use": "sig",
"alg": "RS256",
"kid": "ZhaR-H6Nug-3jEUOjdUvmXP1NIM",
"x5t": "ZhaR-H6Nug-3jEUOjdUvmXP1NIM",
"n": "xCikm91WJOqHsXhnVYbwek_8Yxf22ica5rh1Ov1Ez_WoyOFOi..",
"e": "AQAB",
"X5c: [
"MIIC*jCCAdqAwIBAgIQDupdR+gtaZCyz1j7+XF1DANBgkqhkiG9w.."
]
}
]
}
```

4.  `FlyerIt` prepares the registration request. The registration request is a statement used for registration submission.

```
{
"registation_statment":
"eyJhbGci0iJIUzINiIsInR5cCI6IkpXVCJ9.."""
}
```

The registration request is simply a signed JSON Web Token (JWT) that contains the necessary metadata used for client registration. Figure 5 is a sample registration statement. A JWT consists of three parts: a header, a payload and a signature. The header consists of three major claims, the `"typ"` claim, which represents the type of token (JWT); the `"alg"` claim, which represents the algorithm that is used for the signature (RSA SHA256); and a `"x5c"` claim, which contains the X.509 public key certificate or certificate chain corresponding to the key that is used to sign the JWT. The signature part represents the signature of the encoded header and the encoded payload. The payload contains the claims needed by the authorization server to identify and register the OpenID Client (`FlyerIt`). Other claims are included in the payload; the `"iss"` and `"sub"` claims, which indicate `FlyerIt` is the base URI; the `"aud"` claim, which references `AdvertiseMe`'s endpoint URL; and `"exp"`, which indicates the expiration time for the JWT.

The registration statement serves three major purposes:

*   It provides the digital certificate needed to validate the signature and trust establishment. The digital certificate is included in the header of a JWT.
*   It establishes the client control of a private key. Since the JWT is signed and contains the signature, verifying the signature using the public key certificate provided in the header allows the authorization server to verify that the issuer of the JWT is who it is claiming to be and truly possesses the private key.
*   It contains the necessary metadata for client registration. These metadata are included in the JWT payload.

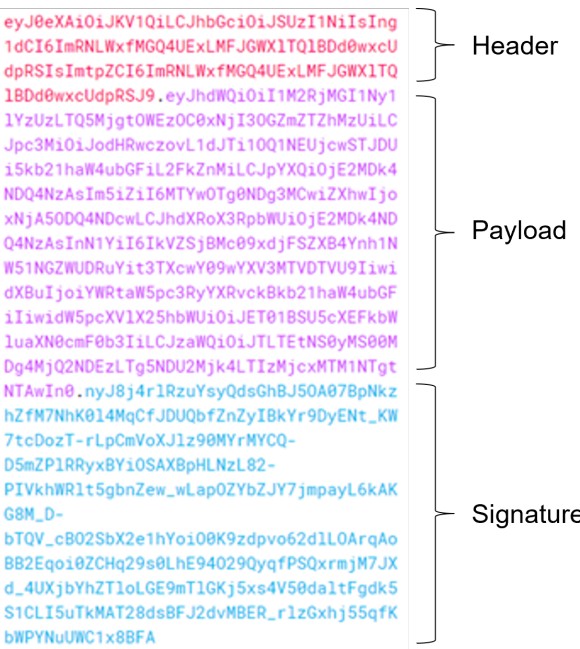

**Figure 5.** Sample Registration JWT.

5.　`AdvertiseMe` validates the registration request through the following steps:

- First, `AdvertiseMe` validates the digital signature of the JWT using the certificate extracted from the JWT header. If the signature cannot be validated, the request is denied. Validating the signature establishes the client control of the certificate's private key.
- Second, a certificate path building and validation is performed. This process is required by the RP to verify the integrity and trustworthiness of all the certificates in a certificate chain that was used to sign a JWT.
- The third step is validating the claims inside the JWT payload. If any of the claims is invalid or missing, the registration request is denied.

6.　If the request is valid, the authorization server (`AdvertiseMe`) returns a response indicating a successful client registration. The registration response should contain a `client_id` claim containing the new client identifier issued by the authorization server.

　　So far, the main concern was to establish a federated identity between a client (`FlyerIt`) and an authorization server (`AdvertiseMe`). Establishing this trust allows end-users to use their identity as a unified identity that serves them in the open world. Now, Bob can utilize his organization identity to access the full services of `FlyerIt` since a trust federation environment was established between `FlyerIt` and `AdvertiseMe`. All users coming from the organization will be able to log in to `FlyerIt` and even achieve this with a single sign-on (SSO) solution. Figure 6 illustrates how `FlyerIt` can sign in Bob, retrieve his basic profile information and allow him to use the services provided.

1.　Bob requests to sign in to `FlyerIt` using his `AdvertiseMe` credentials.
2.　Since `FlyerIt` had previously dynamically discovered and registered with the `AdvertiseMe` identity provider, it redirects Bob to `AdvertiseMe`'s authorization endpoint initiating the authorization code flow. Other grant types might be used; we used the authorization code flow.

```
GET /authorize?
response_type=code&
scope=openid&
state=state_value&
```

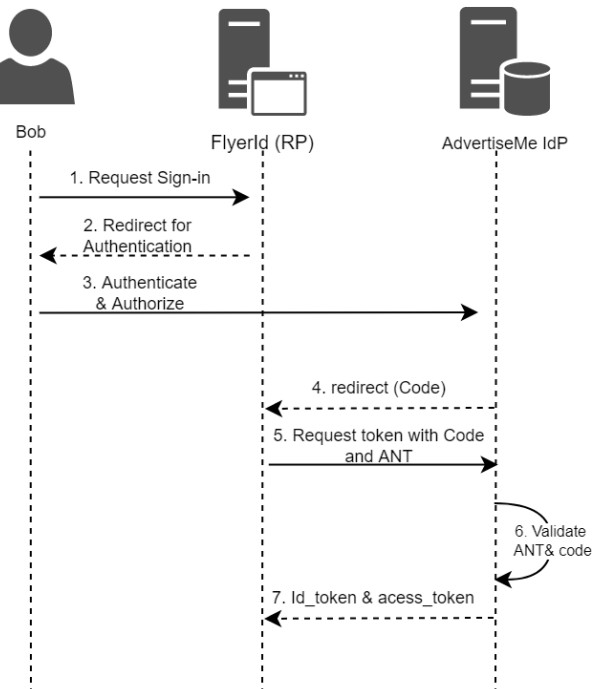

**Figure 6.** Authentication Flow.

```
client_id=registered_client_Id&
redirect_uri=https://flyerit.com/callback1 HTTP/1.1
Host: identity.advertiseme.com
```

3. Bob is redirected to his company's login page to authenticate himself and to authorize `FlyerIt` to access his basic information.

4. When Bob authorizes `FlyerIt` to the requested resources, `AdvertiseMe` returns an authorization code to `FlyerIt` by redirecting Bob to the `redirect_uri` that was presented in the authorization request, which is `https://flyerit.com/callback1` in our example, and returns the authorization code response.

```
HTTP/1.1 302 Found
Location: https://flyerit.com/callback1?
code=authorization_code&
state=state_value
```

5. In the authorization code flow, the client usually submits an authorization code along with his `clientId` and `clientSecret` to the authorization server's token endpoint to receive an `access_token` and an `id_token`. In our case, there was no `clientSecret` assigned to `FlyerIt`; instead, `FlyerIt` prepares an authentication statement similar to the registration statement it prepared for the dynamic registration process. The authentication statement is a signed JWT that contains the following claims:

   - An `"iss"` claim that holds a URI identifying the issuer, which is `FlyerIt` in our case.
   - A `"sub"` claim containing the `clientId`.
   - An `"exp"` claim containing a value for token expiray, and
   - An `"aud"` claim indicating the authorization server (`AdvertiseMe`).

   The authentication statement token should be signed using the same key that was used for the dynamic client registration process. The signed statement serves two purposes: first, it establishes `FlyerIt`'s control of the certificate's private key and second, it provides the digital certificate that is used to validate and verify the signature and establish trust. The following is a token request containing the authentication statement.

```
POST /token HTTP/1.1
Host: identity.advertisme.com
Content-type: application/x-www-form-urlencoded

grant_type=authorization_code&
code=authorization_code&
client_assertion_type=urn:
ietf:params:oauth:client-assertion-type:jwt-bearer&
client_assertion=authentication_statment
```

The `client_assertion_type` in the request body indicates that the client authorization request will provide a signed JWT bearer instead of a `clientSecret`; the `client_assertion` holds the value of the constructed client's authentication statement.

6.  `AdvertiseMe` validates the authentication request. Then, another certificate path validation is performed to make sure the digital certificate provided is valid and can be trusted. In addition, the server validates that the same certificate that were used in the registration process. Lastly, the server validates the provided authorization code, and the claims included in the authentication token. If any of the validation steps fails, `AdvertiseMe` rejects the request.

7.  Upon successful validation, `AdvertiseMe` returns a token response. In addition to the `id_token` and the `access_token` returned by the authorization server, the server can return a `refresh_token` used to refresh the `access_token` for later use.

```
HTTP/1.1 200 OK
Content-type: application/json

{
"access_token": "access_token",
"id_token": "id_token",
"token_type": "Bearer",
"expires_in": 3600
}
```

## 5. Evaluation and Discussion

The primary aim of this evaluation was to capture the effect of establishing a dynamic federated identity model versus the manual steps usually taken by developers to establish a normal federated identity model that aims to scale up their organization to reach multiple business partners. The proposed model was evaluated by conducting a case study where we created an empty web application and implemented the support for five different identity providers establishing identity federation. We chose five of the most implemented OPs. The providers were ADFS and four social providers: Google, Facebook, Twitter and LinkedIn. After the case study, we observed the results by filling out three questions (one of them was an open-ended question) that captured our experience, time and effort to establish a complete end-to-end federation with each IdP individually.

### 5.1. Case Study Setup

Throughout the case study, we started with a sample empty web application, and added IdP's support one by one following these steps: First, we made sure we had user credentials for the specified IdP. Then, we implemented the login support using the specified IdP with the help of any online materials reflecting a real-world scenario for any developer. Lastly, we observed the results and filled out a form that reflected our experience.

We created a web application that asked the user to login with any supported signing-in techniques. If the authentication was successful, the web application said `"Hello, User!"` indicating the name of the authenticated end-user. If the authentication was not successful, the web application showed a message indicating a failed authentication. The application was built using Microsoft's ASP.NET Core SDK with the help of Visual Studio 2022 IDE Community Edition.

*5.2. Results*

To find the effect of establishing a dynamic federated identity model versus the traditional identity federation establishment, we collected some performance parameters, such as time and understandability. We implemented the five IdPs suggested for this case study as a showcase of manually establishing identity federation.

Figure 7 illustrates the time in hours it took to integrate each IdP services to our web application. Similarly, some implementation approaches were easier to understand than others and had relatively simple and easy-to-access documentation online. ADFS was the most complex approach to implement in our opinion, due to the advanced features and the lack of friendly user interface support, where we had to do some `PowerShell` scripting to achieve federation results. On the other hand, Twitter was the simplest to implement knowing that we were more familiar with Google and Facebook.

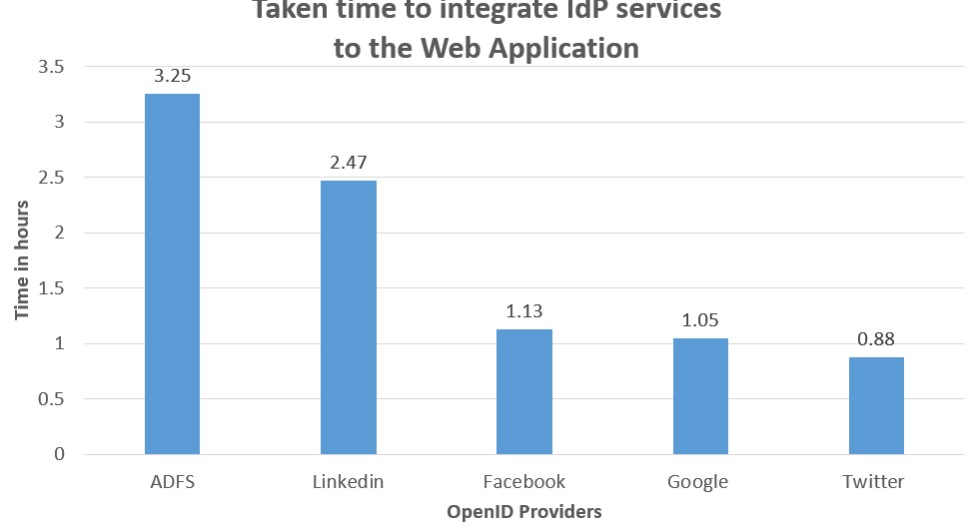

**Figure 7.** Time taken to integrate IdP services to the Web application.

Figure 8 illustrates the simplicity of the integration approaches' metrics, where it shows that Twitter was the simplest approach and Microsoft ADFS was the most complex one.

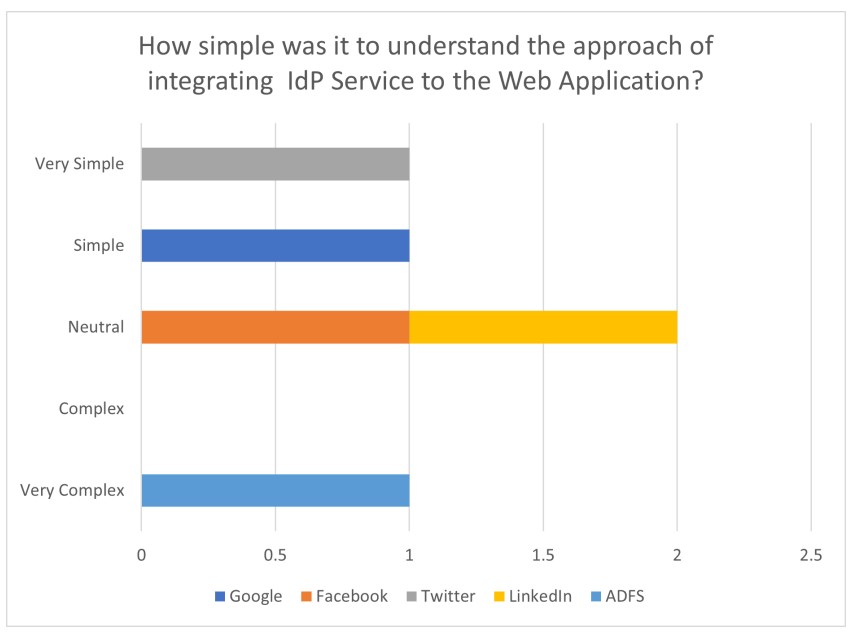

**Figure 8.** How simple was it to understand the approach of integrating IdP Service to the Web application?

The growth of business collaboration platforms has increased the need to collaborate with hundreds of business providers or even more. Through our proposed dynamic identity federation model, organizations can save hundreds of hours by achieving dynamic federation in runtime and serving an enormous number of end-users.

## 6. Conclusions and Future Work

In this study, a dynamic FIM model was proposed based on OIDC, which allowed organizations to establish identity federation dynamically at runtime. By that, manual FIM establishment was alleviated, and the dynamic approach enabled organizational collaborations, removing the barriers on the IT sector. The proposed model consisted of three major steps: The first step was dynamic discovery that was concerned with revealing the location of the OpenID provider (OP) for OpenID relying parties (RPs). The model was based on OIDC's discovery specification [8], which allowed relying parties to detect the location of the OP through `WebFinger` requests in order to be able to utilize OIDC services for end-users. In order for an RP and an OP to initiate OIDC services, the RP needed to be registered as a client at the OP, hence the need for the second step, dynamic client registration. During client registration, a trust model had to be established between both RPs and OPs that allowed them to trust information communicated between each other. The third step was the dynamic trust establishment between OPs and RPs. The proposed model specified how trust could be dynamically established by resolving it from a common trusted third party. The model was designed and implemented in a controlled environment to prove its feasibility. Implementing dynamic FIM can save hundreds of hours and effort for organizations to establish identity federation and to effectively scale up to hundreds of collaborations.

The proposed dynamic identity federation model does not only benefit business organization, but can be extended to health organizations, IoT operations and many other deployment cases. In the case of a health organization, patients can make a great use of identity federations. If Bob was a patient at hospital *A* and decides to go to hospital *B*, the proposed model will allow hospital *B* to establish a trust federation with hospital *A* to access all of Bob's medical records, history and medications records that are stored at hospital *A*. Bob will have a single identity and will be able to use it across all hospitals that support the proposed dynamic identity federation model because it allows them to dynamically establish a trust relationship and start exchanging identity information.

To that end, this study is an initial prototype and proof of concept toward a fully functional browser plugin or application that offers FIM in an easy and transparent way. In addition, we need to do more in terms of usability, scalability and performance testing to see if the proposed model is feasible in the real world. Furthermore, we need to conduct a comprehensive study and threat model analysis for the security and privacy concerns.

**Author Contributions:** A.A.: Conceptualization, Investigation, Validation, Visualization, Methodology, Supervision. N.Y.: Data curation, Formal analysis, Methodology, Software, Writing—original draft. Y.H.: Project administration, Writin—review & editing, Supervision. All authors have read and agreed to the published version of the manuscript.

**Funding:** The study was carried out at Birzeit University with no funding or financial support.

**Data Availability Statement:** All data and implementation steps are included in the manuscript. However, any further details can provided upon request to asadeh@birzeit.edu.

**Conflicts of Interest:** The authors declare no conflict of interest.

**Abbreviations**

The following abbreviations are used in this manuscript:

| | |
|---|---|
| AD | Active Directory |
| ADFS | Active Directory Federation Services |
| API | Application programmable interface |
| CA | Certificate authorities |
| FIM | Federated identity management |
| IdP | Identity provider |
| JWT | JSON Web Token |
| OAuth 2.0 | Open Authentication 2.0 |
| OIDC | OpenID Connect |
| OP | OpenID provider |
| PKI | Public key infrastructure |
| RP | OIDC relying party |
| SAML | Security Assertion Markup Language |
| SP | Service provider |
| SSO | Single sign-On |
| TSP | Trust server provider |
| XRI | Extensible Resource Identifier |

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
