# Peer review of "A Dynamic Federated Identity Management Using OpenID Connect"

_futureinternet, doi:10.3390/fi14110339_

Round 1

Reviewer 1 Report

The work presented in this paper is interesting.

On the other hand, some extra efforts are necessary to provide a clearer context to the readers. Those are:

a) The number of sections is required to review, starting not from 0 but 1. Therefore, also correcting subsections.

b) A final table in the section Related Work is an essential element to provide the general idea of the field.  

c) Section 2 could be called more general as an Experimental Environment. Because set up and environment and the implement could not clearly logical for all.

d) The Evaluation and Discussion section is unacceptable, since it does not provide a detailed appropriated dissertation about both topics. It is a vital section inside the context of the contribution.

e) The Conclusion section requires a special attention. Could be coin as a Conclusions and Future Works. Where a more detailed section may express all efforts during the research and experimental phases. Example is the sentence "The model has been designed and implemented ....", where it will be important to express final comments related to the design and implementation.   

Author Response

Thank you for your valuable comments.

We modified the result section and carried out additional experiments experiment to support our result and conclusion.

We correct the numbering of the sections. In addition, section are renamed.

We provide more details about the evaluation and discussion. In addition, we added a paragraph about the future work.

Moreover, we review the entire manuscript and fix some English language text.

Kindly find the attached file that shows the changes. 

Reviewer 2 Report

I think this paper is ok but some suggestions:

1 add some theoretica analysis of your proposal

2  section 3 is too short, please extend it and adding more discussion

Author Response

Thank you for you positive feedback.

We extend the evaluation and discussion section.

Kindly find the attached file that shows the changes. 

Reviewer 3 Report

This work proposes a dynamic identity federation model to eliminate the manual configuration steps needed to establish organizational identity federation by utilizing the OpenID Connect (OIDC) framework.

The paper is not technically sound and not well organized. In particular, a section on the proposed methodology is missing, and it is not easy to understand the scientific contribution.

The works in the reference section are dated.

It looks like a preliminary draft that needs to be improved. As it stands, the article is not suitable for the journal.

Author Response

Thank you for your comments. We improved the paper by extending some sections and renaming them.

Kindly find the attached file that shows the new changes. 

Reviewer 4 Report

Recommendation: Major Revision

Manuscript ID: futureinternet-1983175

Title: A Dynamic Federated Identity Management Using OpenID Connect

Comments:

In order to solve the issue of organizations while scaling Federated Identity Management (FIM), authors in this paper propose a dynamic identity federation model, which utilizes the OpenID Connect (OIDC) framework. The proposed model consists of three major steps, i.e., the discovery of OpenID service provider, the registration of OpenID relying party and establishing dynamic trust federation. Overall, the quality of this paper should be improved. There are some suggestions given as follows:

1. In the Abstract section, the descriptions on proposed model effectiveness are missing, which are important to the persuasion of this paper.

2. The layout of the paper should be reorganized in the way that sections get titled appropriately.

3. The quality of references should be improved. The relevant papers focus on techniques in the past decade, and these papers should be further enriched. Please incorporate more relevant papers such as Deep reinforcement learning for dynamic computation offloading and resource allocation in cache-assisted mobile edge computing systems; Access Control and Authorization in Smart Homes: A Survey; Multimodal Adaptive Identity-Recognition Algorithm Fused with Gait Perception; Deep reinforcement learning based computation offloading and resource allocation for low-latency fog radio access networks.

4. This paper proposes a model, but there are no chapters describing the proposed model, which is vital for a research paper.

5. The figures in this paper are unprofessional. It is unnecessary to present relevant code in the form of figure.

6. In the experiment part, there are no experimental results presented, comparison approaches and evaluation metrics.

7. What are the advantages of the proposed model in this paper compared with current relevant approaches?

Author Response

Thank you for your valuable comments. We improved the paper by extending some sections and renaming them. Kindly find the attached file that shows the new changes.

Regarding to the suggested references, I could not find the relation between theses references and our paper. They are not appropriate for the topic.

We, represent relevant code as a text in the manuscript.

We added experimental results and evaluation metrics.

Round 2

Reviewer 3 Report

The authors could improve the introduction by including the following works:

DOI: 10.1016/j.patrec.2020.04.038

DOI: 10.1002/cpe.5289

DOI: 10.18293/DMSVIVA2018-002

Finally, the authors should carefully re-read the paper in order to correct typing errors.

Reviewer 4 Report

The revision has addressed the previous concerns well and I recommend it to be accetped.